# The Role of Gene Fusions in Thymic Epithelial Tumors

**DOI:** 10.3390/cancers15235596

**Published:** 2023-11-27

**Authors:** Anja C. Roden

**Affiliations:** Department of Laboratory Medicine & Pathology, Mayo Clinic Rochester, Hilton 11, 200 First St SW, Rochester, MN 55905, USA; roden.anja@mayo.edu

**Keywords:** *MAML2*, *NUTM1*, *EWSR1*, *YAP1*, *NTRK*

## Abstract

**Simple Summary:**

Thymic epithelial tumors (TET) are rare. Therefore, large studies, specifically concerning genetic alterations, are difficult in these tumors and require multi-institutional effort. However, such studies are important because alterations in certain genes of these tumors could be useful in the diagnosis of difficult cases, provide a better understanding of the nature of these tumors, or may even be therapeutically targetable. A few genetic alterations have already been identified in TET. This review will focus on gene fusions that have been reported in a subset of TET. The histopathologic features and clinical characteristics of these subtypes of TET and the potential clinical implications of the gene fusions for outcome of the patient and therapeutic options will be discussed.

**Abstract:**

Thymic epithelial tumors (TET) are rare and large molecular studies are therefore difficult to perform. However, institutional case series and rare multi-institutional studies have identified a number of interesting molecular aberrations in TET, including gene fusions in a subset of these tumors. These gene fusions can aid in the diagnosis, shed light on the pathogenesis of a subset of tumors, and potentially may provide patients with the opportunity to undergo targeted therapy or participation in clinical trials. Gene fusions that have been identified in TET include *MAML2* rearrangements in 50% to 56% of mucoepidermoid carcinomas (*MAML2::CRTC1*), 77% to 100% of metaplastic thymomas (*YAP1::MAML2*), and 6% of B2 and B3 thymomas (*MAML2::KMT2A*); *NUTM1* rearrangements in NUT carcinomas (most commonly *BRD4::NUTM1*); *EWSR1* rearrangement in hyalinizing clear cell carcinoma (*EWSR1::ATF1*); and *NTRK* rearrangement in a thymoma (*EIF4B::NTRK3*). This review focuses on TET in which these fusion genes have been identified, their morphologic, immunophenotypic, and clinical characteristics and potential clinical implications of the fusion genes. Larger, multi-institutional, global studies are needed to further elucidate the molecular characteristics of these rare but sometimes very aggressive tumors in order to optimize patient management, provide patients with the opportunity to undergo targeted therapy and participate in clinical trials, and to elucidate the pathogenesis of these tumors.

## 1. Introduction

With the advantage of increasingly accessible and affordable molecular testing, gene alterations, including gene fusions, are being identified increasingly more often in thymomas and thymic carcinomas. The presence of gene fusions can aid in the diagnosis of neoplasms, provide more insight into the pathogenesis of these tumors, and potentially be useful for individualized management and targeted therapies. Given the paucity of thymic epithelial tumors (TET), large studies are challenging to perform even though multi-institutional studies of molecular alterations in these tumors have been accomplished [1].

In TET, recurring gene fusions have been identified involving mastermind transcriptional coactivator 2 (*MAML2*), Ewing sarcoma breakpoint region 1/EWS RNA binding protein 1 (*EWSR1*), NUT midline carcinoma family member 1 (*NUTM1*), and neurotrophic tyrosine kinase receptor (*NTRK*). *MAML2* gene rearrangements have been detected in mucoepidermoid carcinomas [2,3,4], metaplastic thymomas [5,6,7,8], and occasional B2 and B3 thymomas [9]. Interestingly, fusion partners of *MAML2* vary depending on the subtype of the TET. While *CRCT1* was found to be fused with *MAML2* in mucoepidermoid carcinomas, as in mucoepidermoid carcinomas elsewhere in the body, *YAP1* was identified as the fusion partner of *MAML2* in metaplastic thymomas and *KMT2A* in type B2 and B3 thymomas. *EWSR1* gene rearrangements are present in hyalinizing clear cell carcinomas [10], similar to hyalinizing clear cell carcinomas in the head and neck [11,12] and lung [13,14,15]. Gene fusions involving the *NUTM1* gene are also found in thymic tumors, specifically in NUT carcinomas, which are considered a subtype of thymic carcinomas [16,17]. In addition to these recurring and well-established gene fusions there is a rare report of *NTRK* fusion in a thymoma [18].

Recurring fusion genes have not been reported in thymic neuroendocrine neoplasms [19].

Herein, gene fusions in TET are reviewed and their importance for diagnosis and pathogenesis are outlined. Although currently they are not used as targets for therapy, potential treatment options and investigative studies are discussed. Table 1 summarizes all TET in which a gene fusion has been identified, including the frequency of the gene fusion, morphologic features of the tumor and other diagnostic features.

## 2. *MAML2* Rearrangements

In the thymus, *MAML2* rearrangements were first identified in mucoepidermoid carcinomas [2,3,4]. Subsequently they have also been found in metaplastic thymomas [5,6,7] and in rare B2 and B3 thymomas [9]. Despite the common *MAML2* fusion partner, the corresponding fusion genes vary based on the tumor subtype in which they occur. For instance, in mucoepidermoid carcinomas, the reported fusion gene is cysteine-rich C-terminal 1 (*CRTC1*) leading to *MAML2::CRTC1* fusions [2]. In metaplastic thymomas the fusion partner is Yes associated protein 1 (*YAP1*), leading to *YAP1::MAML2* fusion genes [5,6,8]. In a subset of B2 and B3 thymomas the fusion partner is lysine methyltransferase 2A (*KMT2A*), resulting in *KMT2A::MAML2* fusions [9]. Although these tumors have a common fusion gene, *MAML2*, they are morphologically and pathogenetically different.

Mucoepidermoid carcinomas are characterized by a triad of squamoid, intermediate, and mucus cells (Figure 1A–E) [3]. Their morphologic, immunophenotypic, and molecular features are essentially identical to mucoepidermoid carcinomas elsewhere, such as those of the head and neck, lung, or breast. In the thymus, mucoepidermoid carcinomas are considered a subtype of thymic carcinomas [20]. Here they can also arise in a cyst. *MAML2* rearrangements are identified in 50 to 56% of thymic mucoepidermoid carcinomas [2,3,21]. The presence of *MAML2* rearrangements aids in the diagnosis, specifically in high grade mucoepidermoid carcinomas, and can be utilized to distinguish mucoepidermoid carcinomas from thymic adenosquamous and squamous cell carcinomas as those do not harbor *MAML2* rearrangements [3]. In a study of 20 thymic mucoepidermoid carcinomas, 56% harbored a t(11;19) (q21;p13) *CRTC1::MAML2* gene fusion [2]. *CRTC3::MAML2* [22] or *EWSR1::POU5F1* [23] gene fusions, which have been identified in some head and neck mucoepidermoid carcinoma, have not been reported in thymic mucoepidermoid carcinomas. Furthermore, *YAP1* and *KMT2A* were also not identified as fusion partners in thymic mucoepidermoid carcinomas [2]. Interestingly, the presence of *CRTC1::MAML2* gene fusion in thymic mucoepidermoid carcinomas was associated with classic tumor histology, lower pT and TNM stage, and better overall survival [2].

A meta-analysis of thymic mucoepidermoid carcinoma [21], including 41 such patients from 24 studies, revealed a mean patient age of 49.8 years (range, 8 to 87 years) without any obvious sex predilection. The mean tumor size was 7.6 cm (range, 1.5 to 20 cm). Patients most commonly presented with dyspnea and/or chest pain; about a third of the patients were asymptomatic and the tumor was found incidentally. Various grading systems have been reported for mucoepidermoid carcinoma. In the thymus, the WHO does not recommend any particular grading system; however, in the lung, these tumors are divided into low- and high-grade tumors according to the WHO [20]. In the abovementioned meta-analysis most patients (67%) had low grade morphology with only 27% showing high grade features [21]. The 5 year and 10 year survivals have been reported as 69% and 43%, respectively [2]. Twenty-five and forty-five percent of patients died of disease with a median survival of 40 and 12 months, respectively [2,21]. Worse overall survival was associated with higher pT stage, higher TNM stage, residual tumors, greater tumor size, high grade tumor histology, and absence of *CRTC1::MAML2* fusion [2]. Lymph node sampling was recommended and histologic grade and tumor stage/resectability were suggested as the main prognostic parameters [21]. Indeed, lymph node sampling is recommended in all resection specimens of TET; moreover, in thymic carcinomas which include mucoepidermoid carcinomas, deep thoracic lymph nodes (N2) should be sampled by surgeons [24].

The differential diagnosis of mucoepidermoid carcinomas in the thymic gland includes multiloculated thymic cyst, adenocarcinoma, and adenosquamous carcinoma. These entities do not harbor *MAML2* rearrangements. However, as *MAML2* rearrangements are only identified in about half of mucoepidermoid carcinomas, caution must be practiced in its absence. An invasive growth pattern of glands and epithelioid cells argues against a multiloculated thymic cyst. The presence of p63- or p40-positive tumor cells argues against an adenocarcinoma.

Metaplastic thymomas are very rare neoplasms that only occur in the thymus [25]. They are characterized by a biphasic morphology including a darker component of ovoid, bland appearing epithelial cells with inconspicuous nucleoli and a paler component of fibroblast-like spindle cells with ample pale cytoplasm (Figure 2A,B). The two components are either clearly separated or merged. Mitotic activity is not increased, and necrosis is not identified. The diagnosis in general is based on morphology. Keratin AE1/AE3 highlights the epithelial component and may or may not stain the spindle cell component. P40 marks only the epithelial component and is negative in the spindle cell component. *MAML2* rearrangements have been identified in 77 to 100% of these tumors (Figure 2C,D) [5,6,7,8]. In metaplastic thymomas the fusion partner is *YAP1*, resulting in *YAP1::MAML2* fusion genes fusing exon 1 or exons 1–5 on *YAP1* with exons 2–5 on *MAML2* [5,6]. However, fluorescence in situ hybridization (FISH) studies for *MAML2* rearrangements are usually not necessary for the diagnosis. Moreover, the split signal that results from an intrachromosomal inversion on chromosome 11 in the *YAP1::MAML2* fusion using a break apart FISH may be subtle and can be missed [7]. Recently, expression of YAP1-C terminus by immunohistochemistry has been shown to be highly sensitive and specific for metaplastic thymoma when evaluated in the context of other TET, including type A, AB, B1, B2, and B3 thymomas, and thymic carcinomas [7].

Both the epithelial and spindle cell component of metaplastic thymomas were considered to be of neoplastic nature by at least some authors, although molecular evidence was lacking [26]. More recently, *MAML2* rearrangements have been observed in both the epithelial and spindle cell component (Figure 2C,D) suggesting that both components are at least clonal and likely neoplastic [6]. Mutations in *POLE*, *HRAS*, *ALK*, *CDK4*, *PTEN*, and *BRAF* were also identified in a number of metaplastic thymomas [6,8]. No other genetic alterations, including *GTF2I* mutation, have been found in these neoplasms [5,7,25].

Metaplastic thymomas are indolent tumors that occur at a median age of 50 years (range, 28 to 71 years) with a slight female predominance [7,20]. Reports of sarcomatoid carcinomas arising in metaplastic thymoma suggest that a small subset of these tumors may develop into a high-grade tumor [27,28]. The majority of patients do not develop metastases, recurrence, or mortality from the disease [5,6,7,8,25]. Only one patient has been reported to develop a recurrence and pleural metastases 14 months after resection of the primary tumor [26].

The differential diagnosis of metaplastic thymomas includes type A thymoma. Type A thymomas lack the biphasic appearance and are composed only of bland ovoid or slightly spindle cells without the spindle cell component. Additional tests are usually not necessary for the differential diagnosis. However, the majority (82% to 100%) of type A thymomas harbor a *GTF2I* mutation [8,29,30] which is absent in metaplastic thymomas. In contrast, *YAP1::MAML2* fusions were not identified in type A thymoma or micronodular thymoma with lymphoid stroma, another entity that may be included in differential diagnoses of metaplastic thymomas [6,8]. In addition, at least a subset of type A thymomas has scattered thymocytes which are also absent in metaplastic thymomas. Sarcomatoid carcinoma can also mimic metaplastic thymoma. However, sarcomatoid carcinoma or carcinosarcoma have high-grade morphologies that typically show easily recognizable mitotic activity and necrosis.

Recently, two additional patients with mediastinal tumors with *YAP1::MAML2* rearrangements were reported that followed a more aggressive clinical course [31]. One patient had a 9.8 cm mass invading the pericardium, right lung, and pulmonary artery and vein. A biopsy was interpreted as spindle sarcoma and the patient underwent neoadjuvant chemotherapy followed by resection of the thymic tumor and bilateral lung metastases. Microscopically the tumor was described as biphasic, composed of both epithelioid and spindle cell component; the epithelioid component expressed keratin and p63, the spindle cells were only focally positive for keratin. Targeted DNA/RNA sequencing revealed not only a *YAP1::MAML2* fusion but also a *TERT* promoter mutation. The second patient was incidentally found to have a mediastinal mass which, on biopsy, was diagnosed as rhabdomyosarcoma. A resection specimen revealed poorly differentiated carcinomatous areas which expressed p63 and keratin and a rhabdomyosarcomatous area with expression of desmin. The tumor was interpreted as carcinosarcoma. The tumor had metastasized to a cervical lymph node and the lung. Sequencing revealed *YAP1::MAML2* fusion in both components, a *TP53* mutation in the carcinomatous and deep deletions of *CDKN2A* and *CDKN2B* in the sarcomatous component [31]. These two cases suggest that *YAP1::MAML2* rearrangements may not only be found in pure metaplastic thymomas but also in more aggressive biphasic mediastinal tumors.

Thymomas are malignant neoplasms that occur in the prevascular mediastinum and in general have a more favorable outcome than thymic carcinomas. *GTF2I* mutations are the most prevalent genetic alterations in thymomas which most commonly occur in type A and AB thymomas [1,29,32]. *MAML2* rearrangements have been identified in 6% of B2 and B3 thymoma but not in other thymomas [9]. In these tumors *KMT2A* on exons 8–11 fused with *MAML2* on exon 2 [9]. These tumors had the typical morphology of B2 or B3 thymomas except for one case that also showed foci of thymic carcinoma. Interestingly, a *KMT2A* mutation has also been identified in a thymic papillary adenocarcinoma that occurred synchronously with a type A thymoma [33]. This 53 year-old woman presented with three nodules in the thorax including a 3 cm papillary adenocarcinoma in the anterior superior mediastinum, a 4 cm type A thymoma in the anterior mediastinum near the pericardium and a minimally invasive adenocarcinoma in the right upper lobe lung. Among other molecular alterations the thymic adenocarcinoma harbored a mutation in *KMT2A* c.2155A > C(p.S719R), while the type A thymoma harbored a different *KMT2A* mutation c.5343del(p.V178Yfs*38).

In thymic tumors, *MAML2* rearrangements can aid in the diagnosis, specifically in mucoepidermoid carcinoma. These rearrangements may play a role in the OKpathogenesis, at least in a subset of TET. Studies in mice have confirmed *CRTC1::MAML2* as an oncogenic driver for the development and maintenance of mucoepidermoid carcinomas [34]. While *MAML2* rearrangements are currently not therapeutically targetable, these studies have also revealed EGFR and CDK4/6 inhibitors as potential therapeutic options in these tumors [34].

## 3. *NUTM1* Rearrangements

In the mediastinum *NUTM1* rearrangements are primarily seen in NUT carcinomas. More recently, these rearrangements have also been described in other small round blue cell tumors, although these tumors appear to be even rarer, specifically in the chest.

Nuclear protein in testis (NUT) carcinomas of the thorax are rare. These tumors are considered an aggressive subset of squamous cell carcinomas [16,35]. They are composed of small-to-medium-sized epithelioid cells with a high nuclear-to-cytoplasmic ratio, monotonous, round-to-oval nuclei, and prominent nucleoli (Figure 3A–E and Figure 4A–E) [36]. NUT carcinomas show high mitotic activity and necrosis. Numerous neutrophils can be seen intermixed with tumor cells in a subset of NUT carcinomas (Figure 4A) [37]. Abrupt squamous differentiation is classic for NUT carcinoma (Figure 3B) but is not always seen. Immunohistochemistry for NUT protein shows a characteristic speckled nuclear staining in the tumor cells (Figure 3E and Figure 4D) [36,38]. NUT carcinomas also frequently express markers of squamous differentiation, including p40 (Figure 3C), p63, and keratin 5 or 5/6 [39,40]; a subset of tumors expresses keratins such as AE1/AE3 and/or CAM5.2 [41,42]; and TTF-1 (Figure 3D and Figure 4B), CD34, or synaptophysin (Figure 4C) can also be seen in some of these tumors [40,43]. Rarely CD30, PLAP, and/or SALL4 expression has been reported in NUT carcinomas [44].

NUT protein is encoded by *NUTM1*. The most common molecular alteration, found in 70% to 78% of all NUT carcinomas, is a t(15;19) (q14;p13.1) translocation resulting in the *BRD4::NUTM1* fusion oncogene [43,45,46]. Bromodomain containing protein 4 (*BRD4*) is a double bromodomain-containing gene which is involved in the regulation of cell cycle progression [17]. Less common translocations include *BRD3::NUTM1* and *NSD3::NUTM1* [43,45]. Other fusion partners of *NTM1* are *CHRM5, ZNF532*, and *ZNF592* [35,43,45]. *NUTM1* rearrangements result in the expression of NUT protein which is normally only expressed in spermatids [47]. Immunohistochemistry for NUT reveals the abovementioned nuclear speckled staining pattern which is 100% specific and highly (87%) sensitive for NUT carcinoma in non-germ cell tumors [36,38]. Confirmatory ancillary tests are not necessary for the diagnosis. However, NUT immunohistochemistry may occasionally be negative. If in doubt, additional molecular or cytogenetic tests, such as next-generation sequencing (NGS), reverse transcription polymerase chain reaction (RT-PCR) for *NUTM1::BRD4*, fluorescence in situ hybridization (FISH) for *NUTM1* rearrangement (Figure 4E), or karyotyping, should be performed. The karyotype of NUT carcinomas is usually simple, with the only aberration being the rearrangement involving the *NUTM1* gene [46]. Besides *NUTM1* rearrangements, no other molecular alterations have been reported in NUT carcinomas [45].

NUT carcinomas are highly aggressive neoplasms [48]. They are most commonly identified in the thorax (51% to 55%), specifically the mediastinum, followed by head and neck (40% to 41%) and other sites [45,49]. In the WHO, NUT carcinoma that occurs in the mediastinum is considered a subtype of thymic carcinoma [20]. The median age of patients is 16 to 50 years old (range, 0.1 to 78 years) [45,49]. There is no apparent sex predilection [45]. Most patients (51% to 77%) have metastatic disease at time of presentation with regional lymph node metastases identified in 68% of patients [45,49].

The prognosis of NUT carcinomas is poor [50,51]. The median overall survival is only 6.5 to 7 months, and the disease is usually fatal [45,49]. Evidence suggests that the prognosis may be associated with the primary tumor site and the molecular aberration. Chau NG et al. [45] showed that primary nonthoracic NUT carcinomas with *BRD3::NUTM1* or *NSD3::NUTM1* fusions have the best overall survival with 36.5 months, in contrast with primary nonthoracic NUT carcinoma with *BRD4::NUTM1* fusion, which have a median overall survival of 10 months, and primary thoracic NUT carcinoma with a median overall survival of only 4.4 months [45]. Long-term survivors (3 years) were only identified in the first two groups [45]. Virarkar M et al. [49] also found that patients with primary head and neck NUT carcinomas have a better overall survival (16 months) than patients with pulmonary or other primary NUT carcinomas (6 months).

Chemotherapy is recommended for unresectable or metastatic tumors followed by local therapy [52]. If the tumor is localized, complete resection followed by radiation, possibly with concomitant chemotherapy, appears to be associated with better prognosis. However, many patients with NUT carcinoma present with metastatic disease and therefore are not surgical candidates. Clinical trials using bromodomain and extraterminal (BET) inhibitors, in combination with conventional chemotherapy (NCT05019716; NCT05372640), are currently listed [53].

The differential diagnosis of NUT carcinoma includes poorly differentiated squamous cell carcinoma, small cell carcinoma, large cell neuroendocrine carcinoma, lymphoma, and SMARCA4-deficient undifferentiated tumor (DUT) among other small round blue cell tumors. A NUT immunohistochemical test should be performed as the expression of other immunohistochemical tests, such as keratins, squamous markers, TTF-1, or synaptophysin, can be either lacking or pose pitfalls as described earlier. Moreover, especially in small biopsies and if crush artifact is present, many of the immunohistochemical tests may be negative and therefore one must be careful not to misdiagnose the tumor as small cell carcinoma. Again, a NUT immunostain should be performed in these situations. Given that NUT is normally expressed in spermatids it is not surprising that NUT may be expressed in a subset of germ cell tumors, including in 100% of spermatocytic seminomas, 74% of seminomas, and 100% of hepatoid and 7% of non-hepatoid yolk sac tumors [54]. However, strong and diffuse expression of NUT has been reported only in 71% of spermatocytic seminomas. Embryonal carcinomas were negative for NUT. The authors did not comment on whether NUT staining in germ cell tumors may be more homogeneous in the nucleus rather than speckled as seen in NUT carcinomas. However, morphological and immunophenotypical features should help to distinguish NUT carcinoma from germ cell tumors, particularly seminomas, which can occur in the prevascular mediastinum and may enter the differential diagnosis of NUT carcinomas.

Small round blue cell tumors with *NUTM1* rearrangement (i.e., *NUTM1*-rearranged sarcomas) have also been described. These tumors have fusion partners, such as *CIC*, *ATXN1, BCORL1, MXD1,* and *MX11* [35,43,45,55], that are different from those of NUT carcinomas. NUT sarcomas with features of extraskeletal myxoid chondrosarcoma and a solid pattern harboring an *X::NUTM1* rearrangement [43], an NUT sarcoma with features of myxoid chondrosarcoma with a *MGA::NUTM1* rearrangement [43], and a high-grade small round/epithelioid cell neoplasm with focal dense fibrous matrix and an *MXD4::NUTM1* rearrangement [43] have also been described. These tumors have been reported in the foot (with lung metastasis 4 years later), lung (with liver metastasis at time of presentation), or chest/pleura, respectively [43]. Awareness of these neoplasms together with NUT immunostain and NGS is helpful in their diagnosis.

## 4. *EWSR1* Rearrangements

Hyalinizing clear cell carcinomas (HCCC) with *EWSR1* rearrangement can occur in the thymus and have been included in the 2021 WHO classification [10,20]. These tumors were first reported in the head and neck [11,56,57,58] and later also in the lung [13,15,59,60,61]. For primary thymic HCCC the 2021 WHO classification indicates that clear cell carcinomas without *EWSR1* rearrangement may or may not represent true thymic clear cell carcinoma [20]. HCCC are characterized by cords, trabeculae, and nests of small-to-medium-sized epithelioid cells. The neoplastic cells have clear or eosinophilic cytoplasm and no significant mitotic activity. The cells are in a background of hyalinized fibrosis or myxohyaline stroma. Pools of mucin or glandular structures with mucin may be present. Lymphocytic infiltrates with germinal centers can be focally seen at the rim or within the lesion. HCCC express p40 and p63 [10,15], with a single case reported to be negative for p63 [62]. They are negative for other myoepithelial markers, TTF-1, napsin A, neuroendocrine markers, and CD117 [10,15,62]. While both reported cases of thymic HCCC showed *EWSR1* rearrangement, in one of the two cases NGS revealed a fusion between exon 13 of *EWSR1* and exon 6 of *ATF1* [10]. Indeed, *ATF1* has been reported as the most common fusion partner of *EWSR1* in HCCC in other organs [13,15,59,62], only rarely have *CREM* [61,63,64] with or without *IRF2::NTRK3* [65] been reported.

In the lung, HCCC has a mostly indolent behavior. In thymic HCCC no deaths have been reported, however, the number of reported cases is small [15]. One of the two reported patients with thymic HCCC, presented at Masaoka Koga stage IIB, received adjuvant radiation because of a positive resection margin, developed pulmonary metastases 12 months after resection, pleural and pericardial metastases 22 months after diagnosis and is alive with partial remission 40 months after diagnosis [10]. The other reported patient had no clinical information available.

## 5. *NTRK* Rearrangements

Chromosomal rearrangement of neutrotrophic tyrosine receptor kinase (*NTRK*) have only been identified in 0.3% of solid tumors. Importantly, TRK inhibitors have been shown to lead to tumor responses [18,66]. A rare case of thymoma harboring an *NTRK* fusion has been reported [18]. Specifically, a 50 year-old male was found to have multiple right pleural-based masses and a 5.8 cm mass in the lateral right hemidiaphragm together with multiple masses in the anterior mediastinum. A biopsy of a lung mass was interpreted as thymoma, suggestive of type B3. The patient was clinically stage IV. After two cycles of chemotherapy, side effects were intolerable and NGS was performed. This revealed an *EIF4B::NTRK3* rearrangement in addition to high tumor mutational burden (31 mutation/Mb), *TP53* variant, and *FGFR1* copy number gain. Subsequently the patient was treated with entrectinib, a tyrosine kinase inhibitor selective for TRKA, B, C, ALK, and ROS1 which has been approved by the Food and Drug Administration to target both NTRK and ROS1 [67]. However, 10 months after treatment was initiated the patient experienced myasthenic crisis during which treatment was held. At that time imaging studies showed that the patient had a partial response with 35% reduction in the sum of the longest diameter by RECIST 1.1 criteria.

## 6. Conclusions

Recurring gene rearrangements have been identified in various TET. None of these gene rearrangements are disease-defining as they can occur in various neoplasms of the thymus. Furthermore, these fusions are not specific to TET and have also been identified in counterparts of the disease elsewhere. Nonetheless, some of these rearrangements, in the correct morphologic and immunophenotypic context, can aid in diagnosis. Others, although in only very rare occasions, may provide the opportunity for targeted therapy. For some tumors, these gene rearrangements may provide opportunities for targeted therapy in the future. The diagnosis of these tumors is important, as some of these neoplasms are very aggressive and the correct diagnosis provides the patient with the opportunity to participate in clinical trials. Moreover, these findings could play a role in pathogenesis in a subset of the tumors in the thymus. The fact that some genes that are involved in these rearrangements are recurring in various tumor subtypes may suggest similar pathogenetic mechanisms.

Molecular studies of these rare tumors need to be expanded in multi-institutional, global studies to provide patients with these sometimes quite aggressive neoplasms with access to individualized targeted treatment and management.

## Figures and Tables

**Figure 1 cancers-15-05596-f001:**
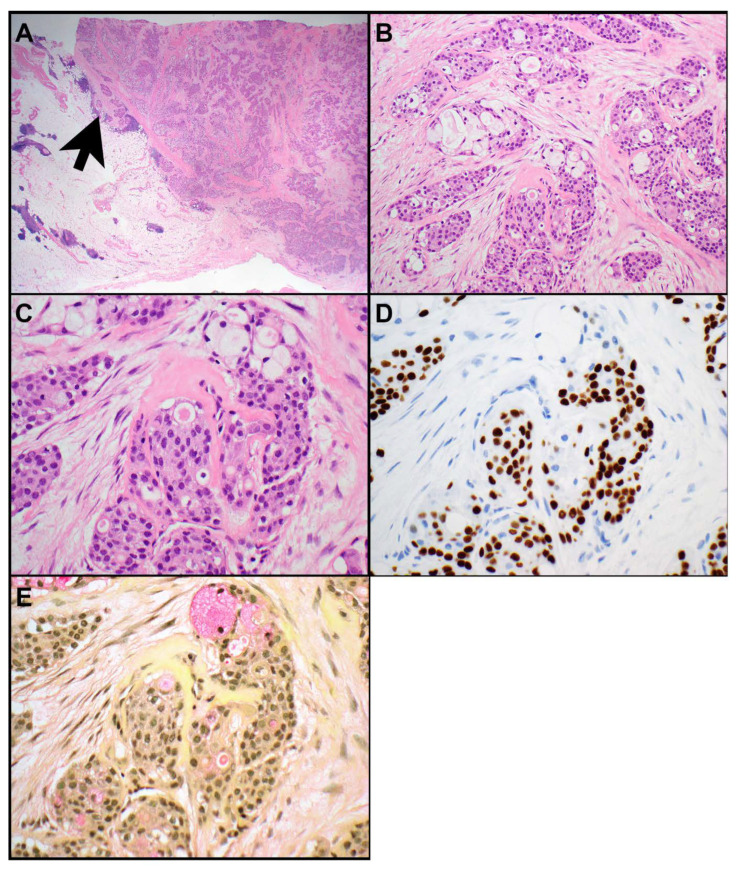
Mucoepidermoid carcinoma. (**A**). This circumscribed neoplasm (right side) is focally invading (arrow) into the surrounding thymic gland tissue (left side). (**B**,**C**). The tumor is composed of squamous and intermediate cells which express p40 (**D**) and of mucus-containing cells. (**E**). Mucicarmine highlights cytoplasmic mucin. Magnification, H&E × 12.5 (**A**), ×200 (**B**), ×400 (**C**), p40 × 400 (**D**), mucicarmine × 400 (**E**).

**Figure 2 cancers-15-05596-f002:**
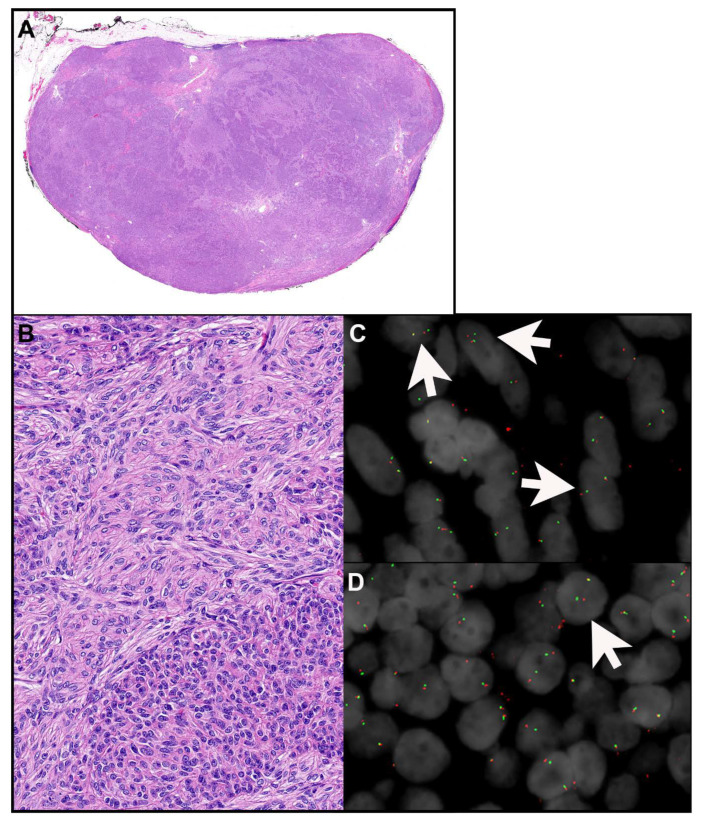
Metaplastic thymoma. (**A**) This well-circumscribed neoplasm shows a biphasic morphology composed of darker and paler areas. (**B**) The paler component (upper part) contains spindle cells with ample pale cytoplasm. The darker component (lower part) is composed of bland, slightly elongated epithelial cells that harbor oval nuclei. (**C**,**D**) Break-apart fluorescence in situ hybridization studies reveal *MAML2* rearrangement using MAML2 3′ (green)/5′ (red) (11q21) probe (arrows point towards cells that show disruption of one red and green signal) in the spindle (**C**) and in the epithelioid (**D**) component. Magnification, H&E x scanned (**A**), ×400 (**B**), *MAML2* FISH × 1000 (oil immersion) (**C**,**D**). Figure 2C,D contributed by Dr. Rhett P. Ketterling, MD and Jamie V. Berg, Mayo Clinic Rochester.

**Figure 3 cancers-15-05596-f003:**
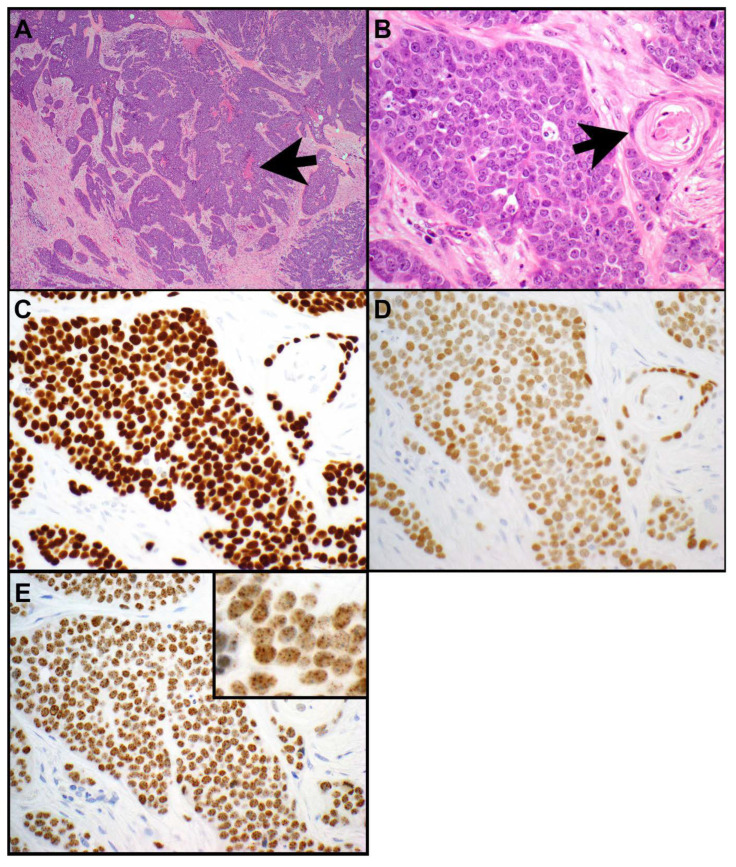
NUT carcinoma. (**A**) Irregular nests of tumor cells are surrounded by a desmoplastic stroma. Areas of necrosis are present (arrow). (**B**) The tumor cells are monotonous, round, with a high nuclear-to-cytoplasmic ratio and prominent nucleoli. High mitotic activity is present. The arrow points towards abrupt squamous differentiation. The neoplastic cells express p40 (**C**), TTF-1 (clone SPT24; weaker expression; note TTF-1 is diffusely expressed by the same population of neoplastic cells that also expresses p40) (**D**), and NUT protein (**E**). The NUT immunohistochemical test shows the characteristic stippled nuclear expression in the tumor cells ((**E**) insert). Magnification, H&E × 40 (**A**), ×400 (**B**), p40 × 400 (**C**), TTF-1 × 400 (**D**), NUT × 400 (**E**), × 600 ((**E**) insert).

**Figure 4 cancers-15-05596-f004:**
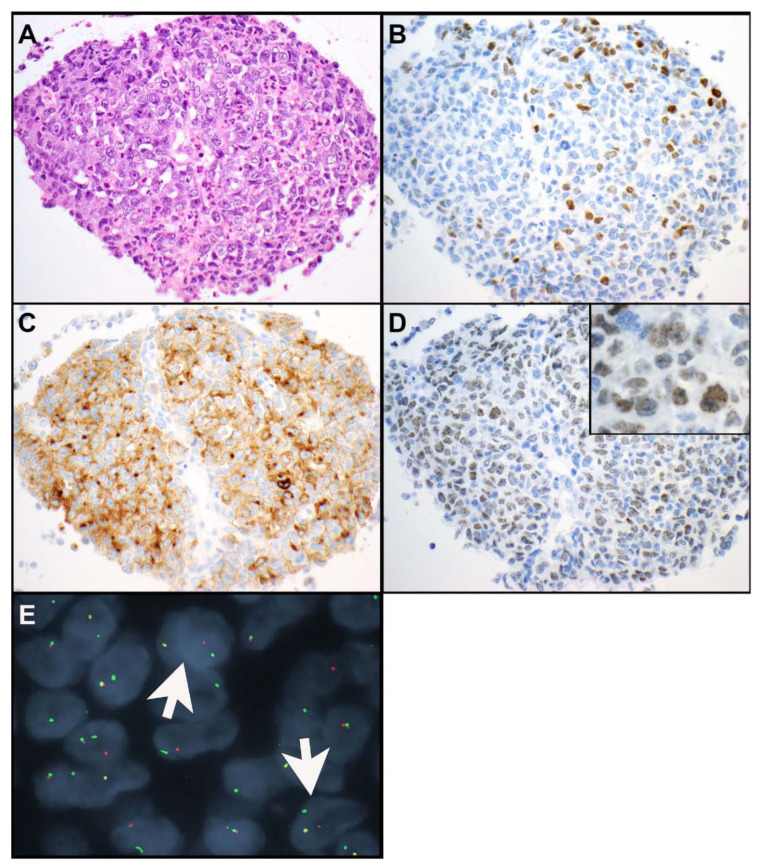
NUT carcinoma. (**A**) Medium-sized neoplastic cells with prominent nucleoli are associated with numerous neutrophils. The neoplastic cells are focally positive for TTF-1 (clone SPT24, staining of scattered tumor cells) (**B**), synaptophysin (**C**), and NUT (**D**). Although NUT shows the characteristic nuclear speckled staining pattern, the overall expression is weak. (**E**). Therefore, break-apart fluorescence in situ hybridization studies using NUTM1 (15q14) 3′ (red)/5′ (green) probe were performed which confirmed *NUTM1* rearrangement (arrows point towards cells that show 1 red, 1 green, and 1 green/red dot, indicating the separation of *5′NUTM1* and *3′NUTM1* signals). Magnification, H&E × 400 (**A**), TTF-1 × 400 (**B**), synaptophysin × 400 (**C**), NUT × 400 (**D**), ×600 ((**D**) insert), FISH *NUTM1* × 1000 (oil immersion). Figure 4E was contributed by Daniel Sill, Mayo Clinic Rochester.

**Table 1 cancers-15-05596-t001:** Morphologic and diagnostic features of thymic epithelial tumors with gene fusions.

Gene	Gene Fusion	Thymic Epithelial Tumor (Frequency of Gene Fusion in %)	Morphologic Features	Other Diagnostic Features
*MAML2*	*MAML2::CRTC1*	Mucoepidermoid carcinoma (50–56)	Triad of squamoid, intermediate, mucus cells	IHC: P63, p40 highlight squamoid and intermediate cells Histochemistry: Mucicarmine highlights cytoplasmic mucin
	*YAP1::MAML2*	Metaplastic thymoma (77–100)	Biphasic with component of bland ovoid epithelial cells and paler spindle cells with ample cytoplasm	IHC: Keratin AE1/AE3 highlights epithelial component and possibly spindle cell component. P40 highlights epithelial component only.
	*MAML2::KMT2A*	Type B2 and B3 thymoma (6)	Type B2 thymoma: Mixture of polygonal neoplastic cells and thymocytes Type B3 thymoma: Predominance of polygonal neoplastic cells with or without a few scattered thymocytes	IHC: Keratin AE1/AE3, p40 highlight neoplastic cells, TdT marks thymocytes in type B2 thymoma and, if present, in type B3 thymoma
*NUTM1*	*BRD4::NUTM1* Other fusion partners: *BRD3, NSD3, CHRM5, ZNF532*, *ZNF592*	NUT carcinoma (70–78, *BRD4::NUTM1*)	Monotonous small-to-medium-sized epithelioid cells with high nuclear-to-cytoplasmic ratio and prominent nucleoli. Abrupt squamous differentiation in subset of cases. Necrosis and high mitotic activity.	IHC: NUT shows speckled nuclear expression; p40, p63, keratin 5 or 5/6 commonly expressed
*EWSR1*	*EWSR1::ATF1*	Hyalinizing clear cell carcinoma (1 case reported)		IHC: p40, p63
*NTRK3*	*EIF4B::NTRK3*	Thymoma (1 case reported)		See above under thymoma

IHC, immunohistochemistry.

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
