# Peer review of "The Role of Gene Fusions in Thymic Epithelial Tumors"

_cancers, 2023, doi:10.3390/cancers15235596_

Round 1
Reviewer 1 Report
Comments and Suggestions for Authors
This is a well written and comperhensive review summarizing the findings on gene fusions in thymic epithelial tumors. It is an actual and important work as most of our knowledge on this subject emerged in the last years. It can be published in the current form.
Author Response
Thank you very much for your kind feedback.
Reviewer 2 Report
Comments and Suggestions for Authors
The discovery of molecular aberration in a subset of rare rare thymic ephitelial tumor is very interesting and can help in definition diagnosis of several subsets. MAML2 rearrangement is discovered in 50% to 56% of muco-epidermoid epithelial thymic carcinoma. It could be a target for future studies in these rare tumors, such as thymoma B2 and B3 and in metaplastic thymoma: in this category YAP1: MAML2 is expressed in 70-100% of cases. There is a need for larger multi-institutional global studies in order to confirm this interesting find. In addition, GTF21 mutation is shown in 82-100% of TET type A thymoma and this is another potential target to new therapies.
Author Response
Thank you very much for your review of the manuscript.
Reviewer 3 Report
Comments and Suggestions for Authors
This is a good and comprehensive review of the medical literature on the topic of gene fusions in thymic tumors.
Author Response
Thank you very much for your review of the manuscript and your kind feedback.
Reviewer 4 Report
Comments and Suggestions for Authors
In this review, the author described characteristics of TET based on fusion genes. The manuscript is well-written and well-organized.
I have one suggestion. To make it more comprehensive especially about diagnosis and treatment, please add these references (PMID: 35970032, PMID: 35820244, PMID: 34161910, PMID: 35970033, PMID: 36497358, PMID: 37190190, PMID: 37190202,   PMID: 36547157, PMID: 36230684) and related descriptions.
Author Response
Thank you very much for your thorough review of the manuscript and your kind feedback.
With all due respect, I would prefer not to include these references. They definitely represent important contributions to treatment options in TET, however, none of these examples represent fusion-targeted therapies.